# Risk factors and outcome due to extended-spectrum β-lactamase-producing uropathogenic *Escherichia coli* in community-onset bloodstream infections: A ten-year cohort study in Sweden

Martin Holmbom[1,2☯]*, Vidar Möller[1☯], Loa Kristinsdottir[1], Maud Nilsson[3], Mamun-Ur Rashid[4], Mats Fredrikson[5], Björn Berglund[3], Åse Östholm Balkhed[1]

1 Division of Infectious Diseases, Department of Biomedical and Clinical Sciences, Faculty of Medicine and Health Sciences, Linköping University, Linköping, Sweden, 2 Department of Urology and Department of Biomedical and Clinical Sciences, Linköping University, Linköping, Sweden, 3 Department of Biomedical and Clinical Sciences, Linköping University, Linköping, Sweden, 4 Department of Health, Medicine and Caring Sciences, Linkoping University, Linkoping, Sweden, 5 Department of Biomedical and Clinical Sciences and Forum Östergötland, Faculty of Medicine and Health Sciences, Linköping, Sweden

☯ These authors contributed equally to this work.
* Martin.Holmbom@regionostergotland.se

## Abstract

### Objective

To study clinical outcome and risk factors associated with extended-spectrum β-lactamase (ESBL)-producing uropathogenic Escherichia coli (UPEC) in community-onset bloodstream infections (CO-BSI).

### Methods

This was a population-based cohort study including patients with pheno- and genotype-matched ESBL-producing *E. coli* and non-ESBL- *E. coli* in urine and blood samples collected in 2009–2018 in southeast Sweden. Seventy-seven episodes of ESBL-UPEC satisfying the inclusion criteria were matched 1:1 with 77 non-ESBL-UPEC for age, gender, and year of culture.

### Results

The most common ST-type and ESBL gene was ST131 (55%), and $bla_{CTX-M-15}$ (47%), respectively. Risk factors for ESBL-UPEC were: previous genitourinary invasive procedure (RR 4.66; p = 0.005) or history of ESBL-producing *E. coli* (RR 12.14; p = 0.024). There was significant difference between ESBL-UPEC and non-ESBL-UPEC regarding time to microbiologically appropriate antibiotic therapy (27:15 h vs. 02:14 h; p = <0.001) and hospital days (9 vs. 5; p = <0.001), but no difference in 30-day mortality (3% vs. 3%; p = >0.999) or sepsis within 36 hours (51% vs. 62%; p = 0.623) was observed.

**Data Availability Statement:** All relevant data are within the manuscript and its Supporting Information files.

**Funding:** This study was supported financially by Alf Grants, Region Östergötland (LIO-793191 and LIO-899871) The funders had no role in study design, data collection and analysis, decision to publish, or preparation of the manuscript.

**Competing interests:** The authors have declared that no competing interests exist.

## Conclusion

The predominant risk factors for ESBL-UPEC were history of ESBL-Ec infection and history of genitourinary invasive procedure. The overall mortality was low and the delay in appropriate antibiotic therapy did not increase the risk for 30-day mortality or risk for sepsis within 36 hours among patients infected with ESBL UPEC. However, these results must be regarded with some degree of caution due to the small sample size.

## Introduction

Urinary tract infection (UTI) is one of the most commonly occurring infections and is often caused by uropathogenic *Escherichia coli* (UPEC). Systemic consequences of UTIs range in severity from asymptomatic bacteriuria to potentially life-threatening bloodstream infections (BSI) [1–5]. Although symptomatic UTI should be controlled with antimicrobial therapy, the increased rate of antibiotic resistance (ABR) among UPEC has become a major concern [6–10]. The incidence of BSI caused by extended-spectrum β-lactamase-producing *E. coli* (ESBL-Ec) has increased worldwide and this pathogen is the most common cause of ABR in clinical isolates in Sweden [11–13]. Furthermore, ESBL-production is associated with resistance to other antibiotic classes including fluoroquinolones, aminoglycosides, and sulphonamides, thus further compromising treatment options for UTIs [14, 15]. ESBL-Ec infections have increased in the community with the spread of ESBL-producing sequence type 131 (ST131). A high prevalence of ST131 (33%) among ESBL-Ec in clinical urine samples in Sweden was shown in a recent study [16]. Another study showed that the risk for infection with ST131 was increased by exposure to cephalosporins or fluoroquinolones during the previous 7 months [17]. Furthermore, in addition to β-lactam resistance, ST131 is associated with high rates of fluoroquinolone resistance [18]. Several studies have described risk factors for ESBL infections such as previous antibiotic treatment, previous hospitalisation, presence of indwelling devices, residency at long-term care facilities, and comorbidity [19–22], and have found high rates of ESBL-Ec-related mortality [23]. However, whether ESBL-Ec infection is associated with increased mortality remains controversial. Some studies have attributed elevated mortality to inappropriate antimicrobial therapy and others to ESBL-production [24–27]. The impact of increased incidence of community-onset BSI (CO-BSI) caused by ESBL-UPEC on the course of the disease and mortality in the Swedish setting remains unclear. The aim of this cohort study was to analyse the epidemiology, clinical outcome, risk factors, and antimicrobial resistance of BSIs caused by ESBL-UPEC.

## Material and methods

### Study design and setting

This was a population-based cohort study on ESBL-Ec BSIs with UTI origin occurring 1st January 2009 to 31st December 2018 in the Swedish county of Östergötland. A nested case-control design was used within the cohort to investigate clinical outcome and risk factors between ESBL-UPEC and non-ESBL-UPEC. Isolates came from all over the county of Östergötland, including a tertiary care university hospital, two general hospitals and a district hospital. The catchment population was 430,000 in December 2009 and 460,000 in December 2018.

## Data collection

From the Clinical Microbiology Laboratory database, the following dataset was collected between 2009 and 2018: all blood and urine isolates with ESBL-Ec and non-ESBL-Ec; year and time of blood culture; and antibiotic susceptibility patterns. The dataset was entered into a second database where it was linked to the patient-administration system providing the following data: gender; age, comorbidity; admitting department; date of admission; date of discharge; and mortality.

To determine clinical outcome, risk factors, and antibiotic resistance associated with ESBL-UPEC in CO-BSI among adults ($\geq$ 18 years), the following inclusion criteria were required: culture-confirmed CO-BSI with ESBL-Ec in blood with pheno- and genotypically matched urine isolates (taken 0–7 days prior the blood culture); and registered and treated in the county of Östergötland during the study period. For non-ESBL UPEC, the same criteria were required with two exceptions; blood and urine isolates were only phenotypically matched, and blood and urine cultures were taken on the same day (**Fig 1**). The medical records were reviewed with a Case Report Form (CRF) for each patient. The following were documented: demographic characteristics; limitation of level of care; nursing home residency; severity of illness (indicated by Sequential Organ Failure Assessment SOFA-score); ICU care within 24 hours; intravenous fluids administration; urinary catheterisation in the emergency department; laboratory data; antimicrobial regimen; clinical outcome; medical history including data on antibiotic use in the past 3 months; previous hospitalisation; UTIs within the previous 12 months; recurrent UTI; proton pump inhibitor exposure; previous findings of ESBL-producing *E. coli*; genitourinary tumour; genitourinary invasive procedure within the previous 12 months; immunosuppressive therapy; and prior invasive procedures or devices.

## Microbiology

Isolates were defined as ESBL if the *E. coli* isolate demonstrated a positive phenotypic test indicating production of classic ESBLs, carbapenemases or the presence of a genetically verified plasmid-carried AmpC-type β-lactamase according to the case definitions of the Swedish Public Health Agency and European Committee for Antimicrobial Susceptibility Testing (EUCAST) recommendations [28, 29]. Results from the routine microbiological laboratory were used for species identification, phenotypic antibiotic susceptibility testing (AST) matching between blood and urine isolates, and phenotypic ESBL detection. All *E. coli* isolated in blood, and all ESBL-producing isolates in blood and urine were frozen and stored at -70 ºC pending culture. Frozen samples were thawed at room temperature and cultivated on blood agar plates. AST for 22 antibiotics (**Fig 2**) was performed in accordance with EUCAST [30] recommendations. Inhibition zone diameters were interpreted using EUCAST breakpoints. Zone diameters classified as within the "area of technical uncertainty" were interpreted as resistant. *E. coli* ATCC 25922 was used as control strain.

## DNA extraction, library preparation and whole-genome sequencing

Total DNA was extracted by using the EZ1 DNA Tissue Kit (Qiagen, Hilden, Germany). A sequencing library for whole-genome sequencing was constructed using 20 ng of DNA. The QIAseq FX DNA Library Kit (Qiagen, Hilden, Germany) was employed for library preparation. The quantity and quality of DNA was gauged with a Qubit 2.0 fluorometer and a QIAxcel instrument (Qiagen, Hilden, Germany), and then paired-end sequenced on a MiSeq instrument (Illumina, San Diego, CA). The average coverage was 73.9.

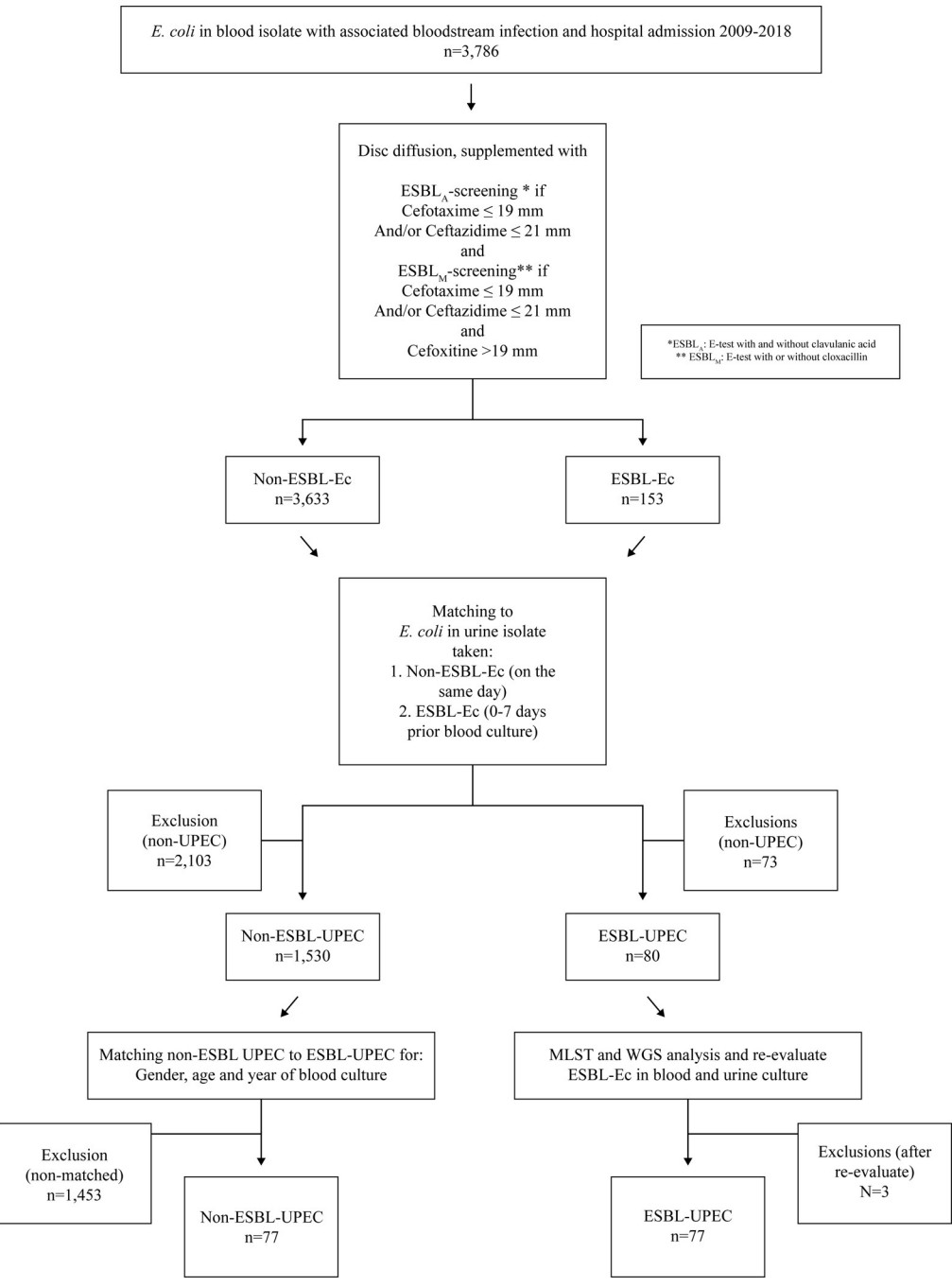

**Fig 1. Flow chart of patients with *E. coli* in blood isolate with associated bloodstream infection, 2009–2018 who were assessed for eligibility (n = 3,786) and further inclusion of ESBL-UPEC (n = 77) and non-ESBL UPEC (n = 77) in the study.**

## Genome assembly and bioinformatic analysis

Genome assembly of the reads obtained from the whole-genome sequenced isolates was performed with CLC Genomics Workbench v.9.5.1 (Qiagen). Multilocus sequence typing (MLST) and identification of antibiotic resistance genes was performed with CLC Genomics

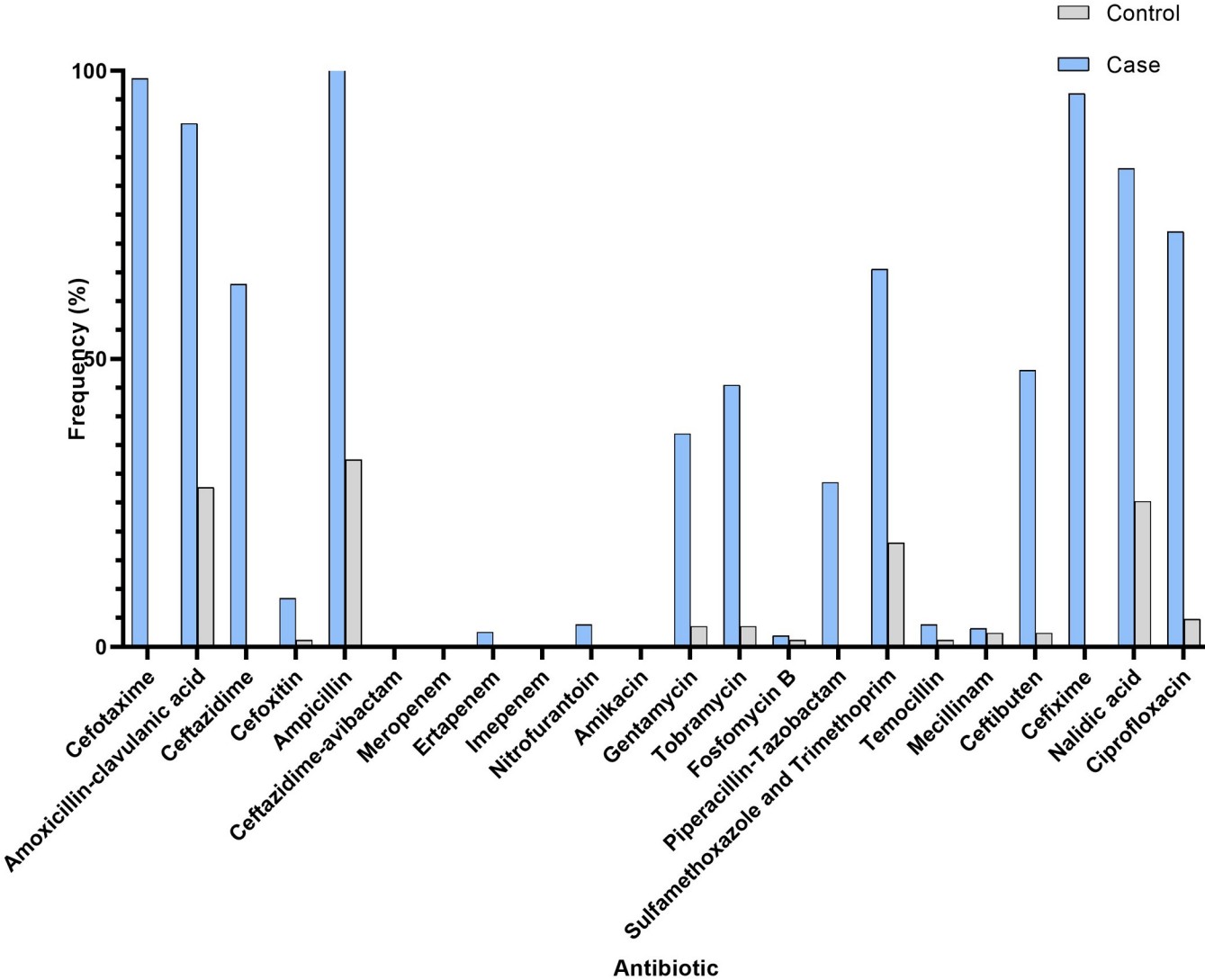

**Fig 2. Antibiotic resistance levels, ESBL-UPEC (case) isolated from blood and urine were matched 1:1 with non-ESBL UPEC (control) for age, gender and year of culture 2009–2018 in southeast of Sweden.** Bars represent frequency of isolates with zone diameters corresponding to resistant or area of technical uncertainty according to EUCAST breakpoints.

Workbench v.9.5.1 (Qiagen, Hilden Germany) by using the database from the Center for Genomic Epidemiology.

## Definitions

Bacteraemia was defined as the finding of ESBL-Ec or non-ESBL-Ec in a blood culture. CO-BSI was defined as a positive blood culture taken on, or within 48 hours of admission. Cultures from readmission within 30 days or patients transferred from another hospital were excluded. Sepsis was defined according to clinical criteria; suspected or demonstrated focus of infection and an acute increase of SOFA score $\geq$ 2 points (a proxy for organ dysfunction) [31]. Empirical antibiotic therapy was defined as initial therapy without AST results of the causative pathogen. Microbiologically appropriate empirical antibiotic therapy was defined according to cultured pathogen and its susceptibility pattern. Antibiotic therapy was defined as appropriate

when an effective antibiotic agent (as determined by *in vitro* susceptibility testing) at the usual recommended dosage was administered [30, 32]. Mortality was defined as all-cause mortality within 30 days. Length of hospital stay was calculated as the time from admission to hospital until discharge.

## Ethical considerations

This study was approved by the Regional Ethics Review Board in Linköping, Sweden. Informed consent was not required. No details of the patients are disclosed and thus patient identity is secure (Ref.no:2011/259-32 and 2017/300-31).

## Statistical analyses

Data are presented as counts and percentages, means and standard deviations (SD) or medians and percentiles (25th-75th). Numerical and categorical variables were assessed using Student´s t test, Fisher´s exact test or $Chi^2$-test. A two-tailed p-value $<0.05$ was considered statistically significant. Significant covariates in univariate analyses were used in a binomial regression model with ESBL-UPEC as dependent variable to investigate possible risk factor for ESBL-U-PEC BSI. The statistics programme SPSS software version 25 was used.

## Results

### Demographic and clinical characteristics

During the study period, a total of 3,786 confirmed cases of *E. coli* BSI (153 ESBL-Ec) were identified. Of these, 76 patients with 77 episodes of ESBL-producing UPEC met the inclusion criteria and were 1:1 matched (by gender, age, and year of culture) to patients with non-ESBL UPEC (n = 77). Demographic and clinical characteristics of the patients with ESBL-UPEC and non-ESBL UPEC are shown in **Table 1**. Comorbidity, severity of disease, laboratory data, and signs of sepsis on admission to hospital did not significantly differ between the groups (**Tables 1** and **S1**). In all, 86 (56%) episodes were admitted to a tertiary care university hospital, 67 (44%) to a general hospital and one to a district hospital.

### Antibiotic resistance

The resistance rates to clinically important antibiotics among ESBL-UPEC were as follows: cefotaxime (99%); ciprofloxacin (71%); trimethoprim-sulfamethoxazole (63%); tobramycin (46%); piperacillin-tazobactam (27%); temocillin (3%); and ertapenem (3%). Corresponding rates for non-ESBL-Ec were: cefotaxime (0%); ciprofloxacin (5%); trimethoprim-sulfamethoxazole (18%); tobramycin (4%); piperacillin-tazobactam (0%); temocillin (1%); and ertapenem (0%). Resistance to meropenem, imipenem, and amikacin was not observed among any isolate. Antibiotic resistance among ESBL-UPEC and non-ESBL UPEC is summarised in **Fig 2**.

### Clonal distribution

Among ESBL-UPEC, ST131 was the most frequently observed ST (54.5%), followed by ST38 (10.4%) and ST405 (9.1%). For non-ESBL-UPEC, the most common ST was ST69 (15%), followed by ST73 (13.8%) and ST95 (12.5) (**S2 Table**).

### Antimicrobial resistance genes

The most common ESBL gene was $bla_{CTX-M-15}$ (47%), followed by $bla_{CTX-M-14}$ (17%) and $bla_{CTX-M-27}$ (16%). All other ESBL-genes were found among 4% or less of the isolates. These

**Table 1. Demographics and baseline characteristics of patients with uropathogenic *Escherichia coli* (UPEC) bloodstream infection[a].**

| | ESBL UPEC (n = 77) | Non-ESBL UPEC (n = 77) | p-value |
|---|---|---|---|
| **Demographics** | | | |
| Gender, male (%) | 52 (68) | 52 (68) | 0.999 |
| Mean age, years, (SD) | 68 (17) | 68 (16) | 0.761 |
| Patient with any limitation of level of care before admission (%) | 4 (5) | 2 (3) | 0.681 |
| Charlson comorbidity index (SD) | 2.6 (2.1) | 3.0 (2.8) | 0.329 |
| Charlson comorbidity index, update (SD) | 2.1 (1.8) | 2.3 (2.3) | 0.638 |
| **Antibiotics in past 12 months** (%) | 51 (66) | 28 (36) | *<0.001* |
| 3rd-generation cephalosporin | 18 (23) | 7 (9) | *0.020* |
| Fluoroquinolone | 15 (19) | 7 (9) | 0.071 |
| Carbapenem | 0 (0) | 0 (0) | - |
| Other antibiotic[b] | 18 (23) | 14 (18) | 0.427 |
| **History of:** (%) | | | |
| Hospitalisation in previous 3 months | 22 (29) | 14 (18) | 0.130 |
| UTI within previous 12 months | 38 (49) | 17 (22) | *0.001* |
| Recurrent UTI | 34 (44) | 9 (12) | *<0.001* |
| Genitourinary tumour | 21 (27) | 6 (8) | *0.003* |
| Genitourinary intervention within the previous 12 months | 30 (39) | 6 (8) | *<0.001* |
| PPI exposure within the previous 6 months | 20 (26) | 20 (26) | 0.999 |
| Previous ESBL-producing *E. coli* | 20 (26) | 1 (1) | *0.002* |
| **Nursing home residency** (%) | 6 (8) | 5 (6) | 0.754 |
| **Immunosuppressive therapy** (%) | 7 (9) | 9 (11) | 0.597 |
| **Prior invasive procedure or devices:** (%) | | | |
| Mechanical ventilation | 0 (0) | 0 (0) | - |
| Central venous catheterisation | 1 (1) | 5 (6) | 0.209 |
| Urinary catheterisation | 20 (26) | 7 (9) | *0.006* |
| **Severity of illness at time of BSI** | | | |
| Sepsis on admission (%) | 35 (45) | 27 (35) | 0.092 |
| Sepsis at 36h (%) | 39 (51) | 48 (62) | 0.623 |
| SOFA-score on admission (SD) | 1.7 (1.6), n = 71 | 1.4 (1.6), n = 76 | 0.235 |
| SOFA score at 36h (SD) | 2.4 (2.1), n = 66 | 2.8 (2.7), n = 76 | 0.301 |
| **ICU care within 24 hours** (%) | 3 (4) | 4 (5) | >0.999 |
| **Intravenous fluids in the ED** (%) | 60 (78) | 60 (77) | 0.878 |
| **Urinary catheterisation in the ED** (%) | 44 (57) | 31 (40) | *0.037* |
| **Time to empirical antibiotic therapy, h, mean** | 03:17, n = 77 | 03:23, n = 77 | 0.858 |
| Median (25th-75th percentiles) | 02:29 (01:15–03:56) | 02:07 (00:50–04:36) | |
| **Time to microbiologically appropriate antibiotic therapy, h, mean** | 32:34, n = 75 | 03:57, n = 77 | *<0.001* |
| Median (25th-75th percentiles) | 27:15 (05:05–54:48) | 02:14 (00:50–04:45) | |
| **Empirical antibiotics** (%) | 77 (100) | 77 (100) | *0.990* |
| 3rd generation cephalosporin | 51 (66) | 52 (68) | 0.864 |
| Fluoroquinolone | 2 (3) | 3 (4) | >0.999 |
| Piperacillin/Tazobactam | 7 (9) | 11 (14) | 0.318 |
| Carbapenem | 14 (18) | 4 (5) | *0.012* |
| Other[c] | 3 (4) | 7 (9) | 0.191 |
| **Microbiologically appropriate empirical antibiotic therapy** (%) | 22 (29) | 74 (96) | *<0.001* |
| **Length of hospital stay,** days (median) (25th-75th percentiles) | 9 (5–13) | 5 (3–7) | *<0.001* |
| **All-cause 30-day mortality** (%) | 2 (3) | 2 (3) | >0.999 |

*(Continued)*

**Table 1.** (Continued)

|  | ESBL UPEC (n = 77) | Non-ESBL UPEC (n = 77) | p-value |
|---|---|---|---|
| **Infection-related mortality** (%) | 2 (3) | 2 (3) | >0.999 |

[a]ESBL-UPEC isolated from blood and urine were matched 1:1 with non-ESBL UPEC for age, gender and year of culture 2009–2018 in southeast of Sweden.

[b]Penicillins + beta-lactamase inhibitors, Nitrofurantoin, Trimethoprim-sulfamethoxazole

[c]Penicillins + beta-lactamase inhibitors, Trimethoprim-sulfamethoxazole, Trimethoprim, Aminoglycosides. Data are presented as nr (%) or mean (SD). Pearson chi$^2$, Fisher's exact test or T-test, as appropriate. P values < 0.05 are shown in italics. Time indications are calculated with median, interquartile 25th to 75th percentile range and Mann–Whitney U. Proton pump inhibitor (PPI).

were $bla_{CTX-M-13}$, $bla_{CTX-M-1}$, $bla_{CTX-M-101}$, $bla_{CTX-M-55}$, $bla_{CTX-M-24}$, $bla_{SHV-12}$, $bla_{DHA-1}$, and $bla_{CMY-2}$. No carbapenemase genes were detected. The distribution of ESBL-genes is shown in **Fig 3**.

## Antibiotic therapy and outcome of patients with ESBL-UPEC versus non-ESBL-UPEC BSI

Median times to empirical antibiotic therapy were similar between patients with ESBL-UPEC BSI and those with non-ESBL UPEC BSI (02:29 h vs. 02:07 h, respectively; p = 0.858). As shown in **Table 1**, third-generation cephalosporins were the most commonly used antibiotics in patients with BSI caused by UPEC in both the ESBL and non-ESBL group (66% vs. 68%; p = 0.864). Patients with ESBL-UPEC BSI were less frequently treated with microbiologically appropriate empirical antibiotic therapy compared to patients with non-ESBL UPEC BSI (29% vs. 99% adequacy, respectively; p = <0.001). Time to microbiologically appropriate antibiotic therapy differed significantly between ESBL-UPEC and non-ESBL UPEC BSIs (27:15 h vs. 02:14 h; p = <0.001). Furthermore, the median length of hospital stay (LOS) for ESBL-UPEC BSI was 9 days compared to 5 days (p = <0.001) for non-ESBL UPEC BSI (**Table 1**).

Almost 3% died within 30 days in this study. **Table 1** shows that there was no significant difference in mortality rate between ESBL-UPEC (2 of 77) and non-ESBL UPEC (2 of 77) BSIs, nor regarding the development of urosepsis within 36 h (51% vs. 62%; p = 0.623).

## Risk factors associated with the development of ESBL-UPEC BSI

Univariate analysis comparing ESBL-UPEC and non-ESBL UPEC BSIs was used to determine risk factors for ESBL-UPEC BSI. There were several significant risk factors: antibiotic treatment during the past 12 months (66% vs 36%); use of 3$^{rd}$ generation cephalosporins within 12 months (23% vs 9%); UTI within the previous 12 months (49% vs 22%); recurrent UTI (44% vs 12%); genitourinary tumour (27% vs 8%); genitourinary invasive procedure within the previous 12 months (39% vs 8%); previous findings of ESBL-producing *E. coli* (26% vs 1%); and previous urinary catheterisation (26% vs 9%). (**Table 1**).

Multivariate analysis showed that genitourinary invasive procedure within the previous 12 months (RR 4.66; p = 0.005) and history of ESBL-producing *E. coli* within the previous 24 months (RR 12.14; p = 0.024) were independent risk factors for the emergence of ESBL-UPEC BSI (**Table 2**).

## Subgroup analysis of ESBL-UPEC—inappropriate vs. appropriate empirical antibiotic therapy

A subgroups analysis of ESBL-UPEC comparing microbiologically inappropriate (n = 55) with appropriate (n = 22) empirical antibiotic therapy was performed. Patient characteristics,

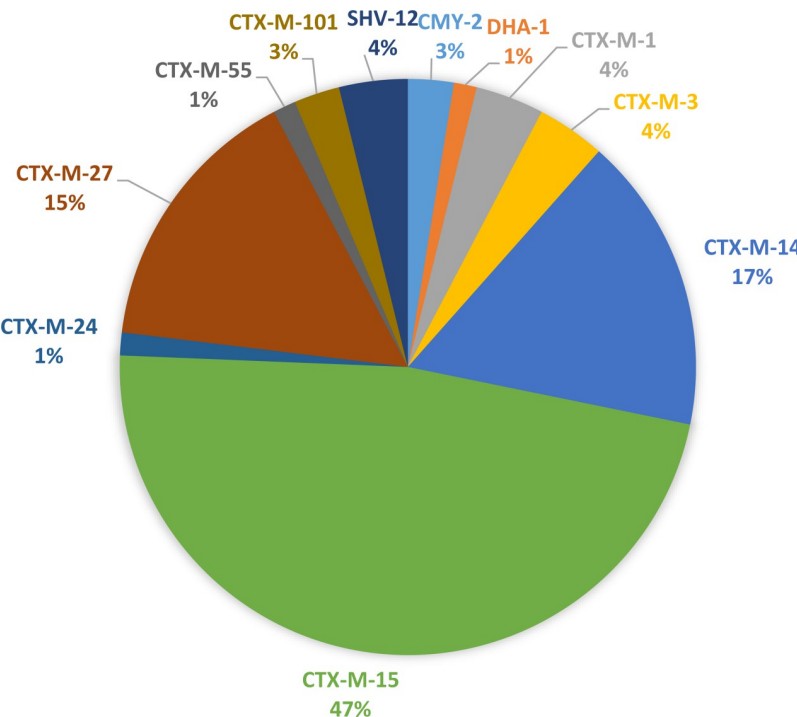

**Fig 3. Distribution of ESBL genes in blood isolates of culture-confirmed community-onset-BSI with ESBL-Ec in blood with pheno- and genotypically matched urine isolates, UPEC (n = 77), taken 0–7 days prior the blood culture.** Registered and treated in southeast of Sweden, 2009–2018.

severity of disease, and comorbidity did not differ between the groups. The patient was significantly more likely to receive appropriate empirical antibiotic therapy if there was a history of previous ESBL within 24 months (50% vs 16%) or UTI within the previous 12 months (68% vs 42%). Median time to microbiologically appropriate antibiotic therapy significantly differed between the groups (47:51 h vs. 03:03 h; p≤0.001). However, 30-day mortality (2% vs. 5%; p = 0.985), SOFA-score at 36 h (2.5 vs. 2.1; p = 0.580) and sepsis at 36 h (53% vs. 46%; p = 0.980) did not differ significantly between the groups (**Table 3**).

**Table 2. Multivariable analysis of risk factors for ESBL-producing UPEC[a].**

|  | RR[*] | 95% CI | p-value |
|---|---|---|---|
| Antibiotics within 12 months | 1.63 | 0.60–4.42 | 0.337 |
| 3rd generation cephalosporin within 12 months | 0.84 | 0.23–3.11 | 0.794 |
| UTI within 12 months | 0.53 | 0.11–2.46 | 0.416 |
| Recurrent UTI | 3.91 | 0.78–19.60 | 0.098 |
| Genitourinary tumour | 2.23 | 0.70–7.13 | 0.176 |
| Genitourinary invasive procedure within 12 months | 4.66 | 1.57–13.79 | *0.005* |
| ESBL-producing *E. coli* within 24 months | 12.14 | 1.39–105.82 | *0.024* |
| Urinary catheterisation | 1.53 | 0.47–4.96 | 0.475 |

[a] Culture-confirmed community-onset-BSI with ESBL-Ec in blood with pheno- and genotypically matched urine isolates (taken 0–7 days prior the blood culture); and registered and treated in the county of Östergötland, 2009–2018
[*]Multivariate binomial regression analysis.

**Table 3. Subgroup analysis of ESBL UPEC (n = 77); inappropriate vs. appropriate empirical antibiotics[a].**

| | Inappropriate empirical antibiotic | Appropriate empirical antibiotic | p-value |
|---|---|---|---|
| | ESBL UPEC (n = 55) | ESBL UPEC (n = 22) | |
| **Gender**, male, n (%) | 38 (68) | 15 (68) | 0.938 |
| **Mean age**, years (SD) | 69 (16) | 65 (16) | 0.307 |
| **History of:** (%) | | | |
| ESBL-producing *E. coli* within 24 months | 9 (16) | 11 (50) | *0.002* |
| Hospitalisation in previous 3 months | 13 (24) | 9 (41) | 0.133 |
| Antibiotics within 3 months | 34 (62) | 17 (77) | 0.200 |
| 3rd generation cephalosporin | 11 (20) | 7 (32) | 0.274 |
| Urinary catheterisation | 11 (20) | 9 (41) | 0.060 |
| UTI within 12 months | 23 (42) | 15 (68) | *0.037* |
| Recurrent UTI | 21 (38) | 12 (55) | 0.055 |
| Genitourinary tumour | 18 (33) | 3 (14) | 0.089 |
| Genitourinary invasive procedure within 12 months | 21 (38) | 9 (41) | 0.825 |
| **Charlson comorbidity index** (SD) | 2.7 (2.1) | 2.6 (2.2) | 0.813 |
| **Charlson comorbidity index, update** (SD) | 2.1 (1.7) | 2.1 (2) | 0.990 |
| **Intravenous fluids in the ED** (%) | 47 (86) | 13 (59) | *0.011* |
| **Urinary catheterisation in the ED** (%) | 32 (58) | 12 (55) | 0.774 |
| **Most common ST-types:** (%) | | | |
| ST131 | 28 (51) | 14 (64) | 0.313 |
| ST38 | 6 (11) | 1 (5) | 0.699 |
| ST405 | 6 (11) | 1 (5) | 0.699 |
| **Severity of illness at time of BSI:** | | | |
| Sepsis on admission (%) | 26 (47), n = 52 | 9 (41), n = 19 | 0.847 |
| Sepsis at 36h (%) | 29 (53), n = 49 | 10 (46), n = 17 | 0.980 |
| SOFA-score on admission (SD) | 1.8 (1.8), n = 52 | 1.3 (1), n = 19 | 0.265 |
| SOFA score at 36h (SD) | 2.5 (2.2), n = 49 | 2.1 (1.7), n = 17 | 0.580 |
| ICU care within 24 hours (%) | 2 (4) | 1 (5) | >0.999 |
| **Time to empirical antibiotic treatment**, h, mean | 03:08 | 03:40 | 0.485 |
| Median (25th-75th percentiles) | 01:45 (01:01–3:56) | 03:03 (02:22–4:12) | |
| **Time to microbiologically appropriate antibiotic therapy**, h, mean | 44:34, n = 53 | 03:40, n = 22 | *<0.001* |
| Median (25th-75th percentiles) | 47:51 (24:03–61:59) | 03:03 (02:22–04:12) | |
| 0–2 hours (%) | 4 (7) | 4 (18) | 0.314 |
| 2–6 hours (%) | 12 (22) | 14 (64) | *0.001* |
| >6 hours (%) | 37 (67) | 3 (14) | *<0.001* |
| **Length of hospital stay**, days, median (25th-75th percentiles) | 9 (6–13) | 9 (5–13) | 0.776 |
| **All-cause 30-day mortality** (%) | 1 (2) | 1 (5) | 0.985 |

[a] Culture-confirmed community-onset-BSI with ESBL-Ec in blood with pheno- and genotypically matched urine isolates (taken 0–7 days prior the blood culture); and registered and treated in the county of Östergötland, 2009–2018. Data are presented as nr (%) or mean (SD). Pearson chi[2], Fisher's exact test or T-test, as appropriate. P values < 0.05 are shown in italics. Time indications are calculated with median, interquartile 25th to 75th percentile range and Mann–Whitney U.

## Discussion

One of the most important reasons for studying risk factors for ESBL-UPEC BSI is the need to identify patients at risk for having this type of infection at the time empirical therapy is initiated. In agreement with previous studies, univariate analysis of ESBL-UPEC BSIs showed that antibiotic use within the past 3 months, history of UTI, and recurrent UTI were significant risk factors. After multivariable correction, we found two dominating risk factors: culture positive for

ESBL-Ec within 24 months, which is in accordance with a previous Swedish study [33]; and history of genitourinary invasive procedure within the previous 12 months. This may indicate that trauma to the epithelium of the urethra and urinary bladder leads to several known risk factors for ESBL development including recurrent UTI with increased use of antibiotics as treatment and increased use of catheters. History of urological disease may explain why patients with ESBL-UPEC received urinary catheterisation in the emergency department to a significantly greater extent than those with non-ESBL-UPEC UTI in this study. Furthermore, ESBL-Ec may persist in the gut in asymptomatic carriers between infections. The ESBL-enzymes found in this study (predominantly CTX-M enzymes) are often plasmid-carried. It is possible that a previous infection with ESBL-producing E. coli predisposes to later ESBL-Ec infection with the same clone due either to plasmid transfer or by new infection with pathogens dormant in the gut.

The all-cause 30-day mortality in this study was only 3%, previous studies in Sweden have shown mortality rates between 6–8% [11, 13, 34] and no difference between ESBL-UPEC and non-ESBL-UPEC BSIs was observed [35, 36]. The differences in mortality in different studies might be due to differences in sample size, severity of illness, primary focus of infection, local epidemiology, healthcare systems, choice and dosing of antibiotics, local empirical treatment recommendations, and whether the infection was community or hospital-acquired. In our study, patients infected by ESBL-UPEC received microbiologically appropriate antibiotic therapy on an average 25 hours later than those infected by non-ESBL-UPEC. Despite this, we did not observe an increased risk for mortality or development of sepsis within 36 hours. Several previous studies investigating the association between inappropriate empirical therapy and mortality and found conflicting results [37–39].

In a previous study [11] by our group, where populations partly overlapped, we found that patients with infections caused by ESBL-UPEC were younger than patients infected with non-ESBL-UPEC (mean age 64 years vs 72 years). Since we matched for age in this study, the mean age in both groups was 68 years, which may contribute to the low overall mortality in this study. Another explanation could be that we selected patients with a urinary tract focus, where source control is easier to achieve, and time to catheterisation may be more important than time to antibiotic treatment.

In this study, ESBL-UPEC BSI was associated with prolonged hospital length of stay. This is a burden on the patient´s health and increases healthcare costs, but this study was not designed to validate ESBL-producing bacteria as a risk factor for prolonged length of stay. There were also some outliers, such as a patient receiving care at a palliative unit for three months, which may influence the outcome.

This study also found the ESBL-UPEC to be a different and more homogenous subgroup of *E. coli* compared to the non-ESBL UPEC strains. The ESBL-UPEC group consisted of 19 STs, of which ST131 constituted 53%, and together the three most common STs (ST131, ST14 and ST27) constituted 80% of the isolates. In contrast, there were 32 different STs in the non-ESBL UPEC group, with no single ST constituting more than 15%.

In our study, $bla_{CTX-M-15}$ was present in nearly half of all ESBL-producing isolates. The second and third most common ESBL genes were $bla_{CTX-M-14}$ and $bla_{CTX-M-27}$. This is in accordance with previous studies both from Sweden and abroad, most likely mirroring the successful expansion of ST131 [19, 22, 40].

Since ST131 was so dominant among ESBL-UPEC in the present study, this will have had a large impact on the results. Further studies are therefore needed to characterise *E. coli* ST131 isolates to identify subclones, phylogenetic distribution, and virulence potential, especially since a low mortality rate was observed in this study-population in Sweden.

Among ESBL-Ec isolates, no resistance to amikacin, meropenem and imipenem was observed, and only low rates of temocillin and ertapenem resistance were observed in our

study. Carbapenems and amikacin are therefore viable treatment options for urosepsis with ESBL-producing pathogens in the setting of this study, while temocillin may be a useful alternative to avoid unnecessary carbapenem use among patients who are not critically ill [41].

## Strengths and limitations

The main strength of this study is that it covers all patients who suffered a BSI caused by ESBL-UPEC in a single county over 10 years. Combined with extensive access to historical, laboratory, and clinical patient data, this is a very comprehensive representation of that cohort.

Another strength of this study is that genuine UPEC cases were studied because of the strict inclusion criteria with both pheno- and genotype matched isolates in blood and urine.

The main limitation of the present study is the small sample size. This may have prevented the detection of minor risk factors for infections caused by ESBL-UPEC. Since this was a retrospective hospital-based study, we could not analyze prehospital risk factors, such as international travel, or causal relationships between risk factors for ESBL UPEC. Furthermore, the mortality figures must be regarded with some degree of caution due to the small sample size. Secondly, our control group of patients with non-ESBL-UPEC BSI comprised a very small non-randomized sample of a large patient group and might not be representative. Another limitation is that blood cultures showing non-ESBL-Ec were only phenotypically matched to urine cultures. This was because the study was retrospective and non-ESBL-Ec isolates from urine are not routinely stored. To minimise the risk that the focus of infection was not the urinary tract in BSI-episodes among the non-ESBL-Ec group, we required that apart from being phenotypically matched, blood and urine samples were to be taken on the same day. Since the main primary focus for *E. coli* BSI is the urinary tract, and with the criteria named above, this limitation is unlikely to have caused selection bias.

The date of previous antibiotic treatment was not registered, only that it was taken within the previous 12 months. Prophylactic antibiotics were not registered.

## Conclusion

The predominant risk factors for ESBL-UPEC BSI were a history of ESBL-Ec infection within the previous 24 months and a history of a genitourinary invasive procedure within the previous 12 months. This study demonstrated an overall low risk for 30-day mortality in ESBL-UPEC CO-BSI and delay in microbiologically appropriate antibiotic therapy did not increase the risk for all-cause 30-day mortality or the risk for developing sepsis within 36 hours after admission for these patients. However, these results must be regarded with some degree of caution due to the small sample size.

## Supporting information

**S1 Table. Laboratory data in emergency department.** ESBL UPEC vs. Non-ESBL UPEC.
(DOCX)

**S2 Table. Distribution of ST-type ESBL UPEC vs. non-ESBL UPEC.**
(DOCX)

## Author Contributions

**Conceptualization:** Martin Holmbom, Vidar Möller, Åse Östholm Balkhed.

**Data curation:** Martin Holmbom, Vidar Möller, Loa Kristinsdottir, Mamun-Ur Rashid, Åse Östholm Balkhed.

**Formal analysis:** Martin Holmbom, Vidar Möller, Mats Fredrikson, Björn Berglund, Åse Östholm Balkhed.

**Funding acquisition:** Björn Berglund, Åse Östholm Balkhed.

**Investigation:** Martin Holmbom, Vidar Möller, Loa Kristinsdottir, Maud Nilsson, Mamun-Ur Rashid, Åse Östholm Balkhed.

**Methodology:** Martin Holmbom, Vidar Möller, Maud Nilsson, Mamun-Ur Rashid, Mats Fredrikson, Björn Berglund, Åse Östholm Balkhed.

**Project administration:** Martin Holmbom, Vidar Möller, Åse Östholm Balkhed.

**Resources:** Åse Östholm Balkhed.

**Software:** Martin Holmbom, Vidar Möller, Mats Fredrikson.

**Supervision:** Mats Fredrikson, Björn Berglund, Åse Östholm Balkhed.

**Validation:** Martin Holmbom, Vidar Möller, Mats Fredrikson, Åse Östholm Balkhed.

**Visualization:** Martin Holmbom, Vidar Möller.

**Writing – original draft:** Martin Holmbom, Vidar Möller.

**Writing – review & editing:** Mats Fredrikson, Björn Berglund, Åse Östholm Balkhed.

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
