## [Decision Letter · Decision Letter 0]

14 Aug 2022

PONE-D-22-16703Risk factors and outcome due to extended-spectrum β-lactamase-producing uropathogenic Escherichia coli in community-onset bloodstream infections: a ten-year cohort study in SwedenPLOS ONE

Dear Dr. Holmbom,

Thank you for submitting your manuscript to PLOS ONE. After careful consideration, we feel that it has merit but does not fully meet PLOS ONE’s publication criteria as it currently stands. Therefore, we invite you to submit a revised version of the manuscript that addresses the points raised during the review process.

We look forward to receiving your revised manuscript.

Kind regards,

Sarah Tschudin-Sutter

Academic Editor

PLOS ONE

Journal Requirements:

Reviewers' comments:

Reviewer's Responses to Questions

**Comments to the Author**

1. Is the manuscript technically sound, and do the data support the conclusions?

Reviewer #1: Yes

Reviewer #2: Yes

2. Has the statistical analysis been performed appropriately and rigorously? 

Reviewer #1: N/A

Reviewer #2: Yes

3. Have the authors made all data underlying the findings in their manuscript fully available?

Reviewer #1: Yes

Reviewer #2: Yes

4. Is the manuscript presented in an intelligible fashion and written in standard English?

Reviewer #1: Yes

Reviewer #2: Yes

5. Review Comments to the Author

Reviewer #1: This is a large population-based cohort identifying the clinical outcomes and risk factors between ESBL-UPEC and non-ESBL-UPEC. It is a well conducted study, and although the authors did not match for co-morbidities (as this might play a role for all-cause mortality), there was no significant difference of Charlson comorbidity index between the two groups

The authors indicate that initial microbiologically appropriate empirical antibiotic therapy has less impact on mortality or risk for development of sepsis among patients with ESBL-UPEC BSI than could be expected, and that this contradicts past studies. although the interpretation is correct, I would not add this message as the first and most important of the study(discussion). Urinary tract infections usually have a good source control, and BSI from UTI with Gram negs can be treated for 7 days only. The authors point these possible biases out for their conclusion. therefore, it would be more appropriate to weight this one conclusion a bit less

it would be interesting to know, whether the use of 3rd generation cephalosporins, which comes with a risk fo ESBL-UPEC, was associated with Genitourinary intervention. in the multivariate analysis, genitourinary intervention is listed as independent risk factor. have prophylactic administrations of antibiotics, such as cephalosporins, before the intervention been taken into account? this is a topic that the authors raise in the discussion part. they however hypothesize that prophylactic use of antibiotics before intervention could have contributed to the risk. is there any possiblity to get the data that for? this would maybe have an implication on prophylaxis.

The authors state that ESBL-producing Enterobacteriaceae within 24months were an independent risk factor for ESBL-UPEC. which enterobacterales do these consist of? were these E. coli? other Enterobacterales? what is the interpretation of the authors for this finding? plasmid transfer?

same applies for patients with recurrent UTIs: was the use of fluoroquinolones/cephalosporins in previous UTI treatment the main risk factor for ESBL-UPEC or does recurrent UTI come as independent risk factor?

minor comments :for better understanding, add "hereof" in the parentesis.

figure 2 needs better resolution

would nice to have a phylogenetic tree for the ESBL escol

Reviewer #2: This population-based cohort study evaluates clinical outcomes and risk factors associated with ESBL-UPEC in community-onset bloodstream infections, along with an analysis of antibiotic resistance. Genitourinary invasive procedures and previous detection of ESBL-producing Enterobacterales were associated with CO-BSI caused by ESBL-UPEC, as well as a longer duration until microbiologically appropriate antibiotic therapy was administered. Despite this, mortality and development of sepsis did not differ between the two groups.

The paper is well written, the results are clearly presented and the findings provide further insight into the epidemiology and burden of ESBL-producing E.coli in a low endemic setting. I have only few minor questions/revisions:

1. In low-ESBL-endemic settings such as Sweden, travel to high endemic areas has frequently been described as a risk factor for ESBL-carriage/infection in previous studies. Despite this being a retrospective study and travel history might not be regularly assessed at hospital admission, it might be of interest whether corresponding information is available in your cohort or why you did not assess travel to high endemic areas in your analyses.

2. In this study, an overall mortality of 3% is reported. Tertiary care centers are considered to comprise a sicker cohort of patients. With regard to the low overall-30-day mortality, it might be interesting to see the proportions of isolates deriving from the tertiary care center, the general hospitals and the district hospital?

3. Length of hospital stay was associated with ESBL-UPEC CO-BSI in univariate analysis, while the subgroup analysis did not reveal a significant association of longer LOS with patients with inappropriate empirical antibiotic therapy. The authors state, that the study was not designed to validate ESBL-producing bacteria as a risk factor for prolonged LOS, so how would you reason this finding?

4. Did you observe any significant changes in distribution of STs of ESBL-E.coli within the study period?

6. PLOS authors have the option to publish the peer review history of their article (what does this mean?). If published, this will include your full peer review and any attached files.

Reviewer #1: No

Reviewer #2: No

---

## [Author Response · Author response to Decision Letter 0]

28 Sep 2022

PONE-D-22-16703 Linköping, Sweden 27th September 2022

Dear Sarah Tschudin-Sutter, Editor

With reference to your decision letter of 14 August 2022

Thank you for the review with all valuable comments on the manuscript "Risk factors and outcome due to extended-spectrum β-lactamase-producing uropathogenic Escherichia coli in community-onset bloodstream infections: a ten-year cohort study in Sweden" and for inviting us to submit a revised version of the manuscript that addresses the points raised in the review process. We have revised the text according to the referees' suggestions and below are our responses to the referees' questions and comments. We have also uploaded the "Revised Manuscript (with Track Changes) and the Manuscript (without Track Changes). 

Please let us know if there is anything in our response letter that needs further clarification.

Yours faithfully

......................................................................................

Martin Holmbom, MD, PhD

Reviewer #1

Reviewer #1: This is a large population-based cohort identifying the clinical outcomes and risk factors between ESBL-UPEC and non-ESBL-UPEC. It is a well conducted study, and although the authors did not match for co-morbidities (as this might play a role for all-cause mortality), there was no significant difference of Charlson comorbidity index between the two groups

AU: Thank you, below are our replies to your questions and comments.

1. The authors indicate that initial microbiologically appropriate empirical antibiotic therapy has less impact on mortality or risk for development of sepsis among patients with ESBL-UPEC BSI than could be expected, and that this contradicts past studies. although the interpretation is correct, I would not add this message as the first and most important of the study(discussion). Urinary tract infections usually have a good source control, and BSI from UTI with Gram negs can be treated for 7 days only. The authors point these possible biases out for their conclusion. therefore, it would be more appropriate to weight this one conclusion a bit less

AU: Thank you, we agree and weight down mortality. We have revised the abstract, discussion, and conclusion accordingly and added a limitation.

Abstract, Lines 29-33 now reads:

“The predominant risk factors for ESBL-UPEC were history of ESBL-Ec infection and history of genitourinary invasive procedure. The overall mortality was low and the delay in appropriate antibiotic therapy did not increase the risk for 30-day mortality or risk for sepsis within 36 hours among patients infected with ESBL UPEC. However, these results must be regarded with some degree of caution due to the small sample size”. 

Discussion

The section regarding mortality has been shifted down and is not stressed as the first and most important result of the study, according to proposal (please see the document “main document – copy) and we have added a limitation see Lines 333-334:

…” Furthermore, the mortality figures must be regarded with some degree of caution due to the small sample size”.

And we have deleted the following in the discussion (see Lines 312-316 in Revised Manuscript with Track Changes):

“Our results indicate that initial microbiologically appropriate empirical antibiotic therapy has less impact on mortality or risk for development of sepsis among patients with ESBL-UPEC BSI than could be expected. These results do not support the conclusions reached by several previous studies, showing that time to appropriate antibiotics is crucial. However, a Dutch study from 2015 showed that inappropriate therapy within 12-24 hours was not associated with a higher 30-day mortality.”

and added a Swedish study from 2022 to the reference list, showing that inappropriate therapy within 12 hours was not associated with higher 30-day mortality in overall BSI.

Heuverswyn et al “Association between time to appropriate antimicrobial treatment and 30-day mortality in patients with bloodstream infections: a retrospective cohort study”, 06 September 2022

Conclusion, Lines 347-353 now reads:

The predominant risk factors for ESBL-UPEC BSI were a history of ESBL-Ec infection within the previous 24 months and a history of a genitourinary invasive procedure within the previous 12 months. This study demonstrated an overall low risk for 30-day mortality in ESBL-UPEC CO-BSI and delay in microbiologically appropriate antibiotic therapy did not increase the risk for all-cause 30-day mortality or the risk for developing sepsis within 36 hours after admission for these patients. However, these results must be regarded with some degree of caution due to the small sample size.

2. It would be interesting to know, whether the use of 3rd generation cephalosporins, which comes with a risk fo ESBL-UPEC, was associated with Genitourinary intervention. in the multivariate analysis, genitourinary intervention is listed as independent risk factor. have prophylactic administrations of antibiotics, such as cephalosporins, before the intervention been taken into account? this is a topic that the authors raise in the discussion part. they however hypothesize that prophylactic use of antibiotics before intervention could have contributed to the risk. is there any possiblity to get the data that for? this would maybe have an implication on prophylaxis.

AU: We agree, it would have been interesting to look at association between genitourinary intervention and 3rd generation cephalosporins. In the multivariate analysis these two variables were included, and only genitourinary intervention was significant indicating that 3rd generation cephalosporin ranks lower than genitourinary intervention as a risk factor. We can only speculate that there is an association between urological interventions and 3rd generation cephalosporin use in this cohort since cephalosporin is a common empirical treatment option in Sweden when a urological infection is suspected, such as after a genitourinary intervention. We did not record the date when previous antibiotic treatment was given, only that antibiotic had been taken within the previous 12 months. Furthermore, the study was not designed to analyze causal relationships between risk factors. As a result, we could not analyze association between genitourinary intervention and use of 3rd generation cephalosporin. 

In the discussion we hypothesize that prophylactic use of antibiotics before intervention could have contributed to the risk for ESBL development. We have deleted this line, because it may confuse the reader and we have no supporting evidence for this speculation. Prophylactic antibiotics were not registered in this study, so we cannot analyze prophylactic use of antibiotics before intervention using our data-set. 

We have now referred to this limitation, see Lines 343-344: 

The date of previous antibiotic treatment was not registered, only that it was taken within the previous 12 months. Prophylactic antibiotics were not registered. 

and added the following to limitations, see Lines 329-332:

The main limitation of the present study is the small sample size. This may have prevented the detection of minor risk factors for infections caused by ESBL-UPEC. Since this was a retrospective hospital-based study, we could not analyze prehospital risk factors, such as international travel, or causal relationships between risk factors for ESBL UPEC.

And deleted the following in the discussion

 “…and as prophylaxis before genitourinary procedure…”

3. The authors state that ESBL-producing Enterobacteriaceae within 24months were an independent risk factor for ESBL-UPEC. which enterobacterales do these consist of? were these E. coli? other Enterobacterales? what is the interpretation of the authors for this finding? plasmid transfer?

AU: The previous cultures showed Escherichia coli only (Klebsiella pneumoniae were ignored, for instance). We believe that recurring infection caused by the same clone carried asymptomatically in the gut by the patient, as well as plasmid transfer could explain this. In a previous study 2008-2016, with partly overlapping populations that included all bloodstream infections (not limited to uropathogens ), we found 25 ESBL-producing Klebsiella pneumoniae. There were 147 cases of ESBL-producing Enterobacterales in that study. When designing the present study, we assumed we would only find a small number of ESBL-producing K. pneumoniae and decided, therefore, that it was more prudent to focus on ESBL-producing E. coli. 

For clarification, the term “ESBL-producing Enterobacteriaceae/bacteria” has been changed to “ESBL-producing E. coli” throughout the manuscript. 

and the following has been added in the discussion, see Lines 275-279:

…“Furthermore, ESBL-Ec may persist in the gut in asymptomatic carriers between infections. The ESBL-enzymes found in this study (predominantly CTX-M enzymes) are often plasmid-carried. It is possible that a previous infection with ESBL-producing E. coli predisposes to later ESBL-Ec infection with the same clone due either to plasmid transfer or by new infection with pathogens dormant in the gut.”

4. Same applies for patients with recurrent UTIs: was the use of fluoroquinolones/cephalosporins in previous UTI treatment the main risk factor for ESBL-UPEC or does recurrent UTI come as independent risk factor?

AU: This is a very interesting comment. There is always the possibility of risk factors affecting each other, and sometimes it is not clear whether a risk factor is the consequence or cause of infection, or whether an association is the result of confounding. A different study design would be needed to establish causality in this study. However, in the multivariable analysis of risk factors for ESBL-UPEC, neither was an independent risk factor.

We have now referred to this limitation, see Lines 329-332: 

… “The main limitation of the present study is the small sample size. This may have prevented the detection of minor risk factors for infection caused by ESBL-UPEC. Since this was a retrospective hospital-based study, we could not analyze prehospital risk factors, such as international travel, or causal relationships between risk factors for ESBL UPEC”.

minor comments :

5. For better understanding, add "hereof" in the parentesis.

AU: We cannot find the locationof this parentesis.

6. Figure 2 needs better resolution

AU: This has been done.

7. Would nice to have a phylogenetic tree for the ESBL escol

AU: Single nucleotide polymorphism analysis was not performed as part of the whole genome sequencing analysis. As a result of this, we cannot create a phylogenetic tree for the ESBL-Ec, but we agree, it would have been interesting.

Reviewer #2

Reviewer #2: This population-based cohort study evaluates clinical outcomes and risk factors associated with ESBL-UPEC in community-onset bloodstream infections, along with an analysis of antibiotic resistance. Genitourinary invasive procedures and previous detection of ESBL-producing Enterobacterales were associated with CO-BSI caused by ESBL-UPEC, as well as a longer duration until microbiologically appropriate antibiotic therapy was administered. Despite this, mortality and development of sepsis did not differ between the two groups.

The paper is well written, the results are clearly presented and the findings provide further insight into the epidemiology and burden of ESBL-producing E. coli in a low endemic setting. I have only few minor questions/revisions:

AU: Thank you, below are our replies to your questions and comments.

1. In low-ESBL-endemic settings such as Sweden, travel to high endemic areas has frequently been described as a risk factor for ESBL-carriage/infection in previous studies. Despite this being a retrospective study and travel history might not be regularly assessed at hospital admission, it might be of interest whether corresponding information is available in your cohort or why you did not assess travel to high endemic areas in your analyses.

AU: We agree, it would have been interesting to look at several prehospital risk factors such as travel abroad etc. But this was a retrospective hospital-based study, and our source of information was hospital medical records and data from the microbiology laboratory, where prehospital information is seldom recorded. We did not have resources to complement the investigation with patient interviews or other source of information. 

Since this could not be handled in our dataset, we refer to this limitation, see Lines 330-332: 

“…since this was a retrospective hospital-based study, we could not analyze prehospital risk factors such as international travel…”

2. In this study, an overall mortality of 3% is reported. Tertiary care centers are considered to comprise a sicker cohort of patients. With regard to the low overall-30-day mortality, it might be interesting to see the proportions of isolates deriving from the tertiary care center, the general hospitals and the district hospital?

AU: Thank you, we have added the following to the results section” Demographic and clinical characteristics”, see Lines 188-189:

”In all, 86 (56%) episodes were admitted to a tertiary care university hospital, 67 (44%) to a general hospital, and one to a district hospital.”

3. Length of hospital stay was associated with ESBL-UPEC CO-BSI in univariate analysis, while the subgroup analysis did not reveal a significant association of longer LOS with patients with inappropriate empirical antibiotic therapy. The authors state, that the study was not designed to validate ESBL-producing bacteria as a risk factor for prolonged LOS, so how would you reason this finding?

AU: This is an interesting question. There was a significant difference in LOS between ESBL UPEC (n77) and non-ESBL UPEC (n77) cases, but there was no significant difference between ESBL UPEC (n55) cases that received appropriate empirical antibiotic therapy compared to ESBL UPEC (n22) cases that received inadequate therapy regarding length of hospital stay. This was unexpected, but could be explained by the small study-population. Furthermore, we did not collect data allowing us to adjust for other (unrelated) reasons for prolonged hospital stay such as comorbidities, time taken for home adaptation, time waiting for a place in a nursing home etc. Even if we found that ESBL-UPEC was associated with prolonged LOS compared to non-ESBL, we would not have been able to adjust for other factors that affect LOS. 

4. Did you observe any significant changes in distribution of STs of ESBL-E. coli within the study period?

AU: The number of isolates each year is too small to make any statistical comparison. There appears not to be a change in the distribution of STs, only that the number of isolates per year increased over the study period, which is in line with Swedish statistics on ESBL prevalence.

---

## [Editor Report · Decision Letter 1]

19 Oct 2022

Risk factors and outcome due to extended-spectrum β-lactamase-producing uropathogenic Escherichia coli in community-onset bloodstream infections: a ten-year cohort study in Sweden

PONE-D-22-16703R1

Dear Dr. Holmbom,

We’re pleased to inform you that your manuscript has been judged scientifically suitable for publication and will be formally accepted for publication once it meets all outstanding technical requirements.

Kind regards,

Sarah Tschudin-Sutter

Academic Editor

PLOS ONE
---

## [Editor Report · Acceptance letter]

25 Oct 2022

PONE-D-22-16703R1 

Risk factors and outcome due to extended-spectrum β-lactamase-producing uropathogenic *Escherichia coli* in community-onset bloodstream infections: a ten-year cohort study in Sweden 

Dear Dr. Holmbom:

I'm pleased to inform you that your manuscript has been deemed suitable for publication in PLOS ONE. Congratulations! Your manuscript is now with our production department. 

Kind regards, 

on behalf of

Dr. Sarah Tschudin-Sutter 

Academic Editor

PLOS ONE